METHODS AND PROTOCOLS

# ESKAPE Act Plus: Pathway Activation Analysis for Bacterial Pathogens

Katja Koeppen,[a] Thomas H. Hampton,[a] Samuel L. Neff,[a] Bruce A. Stanton[a]

[a]Department of Microbiology and Immunology, Geisel School of Medicine at Dartmouth, Hanover, New Hampshire, USA

**ABSTRACT** The last 20 years have witnessed an explosion in publicly available gene expression and proteomic data and new tools to help researchers analyze these data. Tools typically include statistical approaches to identify differential expression, integrate prior knowledge, visualize results, and suggest how differential expression relates to changes in phenotype. Here, we provide a simple web-based tool that bridges some of the gaps between the functionality available to those studying eukaryotes and those studying prokaryotes. Specifically, our Shiny web application ESKAPE Act PLUS allows researchers to upload results of high-throughput bacterial gene or protein expression experiments from 13 species, including the six ESKAPE pathogens, to our system and receive (i) an analysis of which KEGG pathways or GO terms are significantly activated or repressed, (ii) visual representations of the magnitude of activation or repression in each category, and (iii) detailed diagrams showing known relationships between genes in each regulated KEGG pathway and fold changes of individual genes. Importantly, our statistical approach does not require users to identify which genes or proteins are differentially expressed. ESKAPE Act PLUS provides high-quality statistics and graphical representations not available using other web-based systems to assess whether prokaryotic biological functions are activated or repressed by experimental conditions. To our knowledge, ESKAPE Act PLUS is the first application that provides pathway activation analysis and pathway-level visualization of gene or protein expression for prokaryotes.

**IMPORTANCE** ESKAPE pathogens are bacteria of concern because they develop antibiotic resistance and can cause life-threatening infections, particularly in more susceptible immunocompromised people. ESKAPE Act PLUS is a user-friendly web application that will advance research on ESKAPE and other pathogens commonly studied by the biomedical community by allowing scientists to infer biological phenotypes from the results from high-throughput bacterial gene or protein expression experiments. ESKAPE Act PLUS currently supports analysis of 23 strains of bacteria from 13 species and can also be used to re-analyze publicly available data to generate new findings and hypotheses for follow-up experiments.

**KEYWORDS** Prokaryotes, bacteria, ESKAPE pathogens, pathway activation analysis, KEGG, GO terms, Shiny web application, ESKAPE, Shiny, gene ontology, pathogens, prokaryotes, web application

Address correspondence to Katja Koeppen, Koeppen.Katja@gmail.com.

The authors declare no conflict of interest.

The group of bacteria collectively known as ESKAPE (*Enterococcus faecium*, *Staphylococcus aureus*, *Klebsiella pneumoniae*, *Acinetobacter baumannii*, *Pseudomonas aeruginosa* and *Enterobacter* sp.) pathogens are of high medical concern due to their virulence and ability to develop antibiotic resistance (1, 2). Collectively, they are the major cause of life-threatening hospital-acquired infections worldwide (3). As a result, biomedical researchers conduct an increasing number of high-throughput experiments involving these pathogens, generating large data sets from RNA-seq or proteomics experiments that are difficult to interpret without advanced computational skills. The ESKAPE Act PLUS web application enables users to upload

results of gene or protein expression experiments from ESKAPE pathogens and several other strains of bacteria and provides a complete analysis of which Kyoto Encyclopedia of Genes and Genomes (KEGG) pathways (4) or gene ontology (GO) terms (5) appear to be systematically activated or repressed. No programming or other special skills are required. ESKAPE Act PLUS is implemented as an R Shiny standalone application that runs in a web browser and can be accessed freely and openly at http://scangeo.dartmouth.edu/ESKAPE/.

ESKAPE Act PLUS has several advantages compared to publicly available tools that perform gene set enrichment or overrepresentation analysis for biological pathways such as KEGG pathways or GO terms. First, overrepresentation analysis typically requires a criterion such as an arbitrary threshold *P*-value to identify a subset of differentially expressed genes or proteins. According to this criterion, sets of differentially expressed genes can then be classified as "enriched" in genes performing specific biological functions if the subset of differentially expressed genes contains a higher proportion of genes performing these functions than one would expect by chance. Enrichment therefore suggests an association between experimental conditions and biological function. However, selecting different *P*-value cutoffs can change enrichment results, and typical choices such as FDR $<$ 0.05 may result in very small gene sets, limiting power. Second, overrepresentation analysis does not consider the direction of the fold changes (positive or negative) and thus does not predict the biological effect associated with the induction or repression of a given pathway or GO term. By contrast, ESKAPE Act PLUS uses all genes or proteins that were detected in a high-throughput experiment and their associated fold changes to predict overall activation or repression at the level of a biological pathway or GO term based on the fold changes of all genes within a given pathway or GO term. While there are tools such as the commercially available Ingenuity Pathway Analysis that perform pathway activation analysis for humans and other eukaryotic model organisms (6), to our knowledge ESKAPE Act PLUS is the first application that extends this functionality to prokaryotes.

## RESULTS AND DISCUSSION

**Comparison of ESKAPE Act PLUS to existing tools (gap analysis).** ESKAPE Act PLUS is a user-friendly Shiny web application that enables researchers working with prokaryotes to interpret high-throughput experiments such as RNA-seq and proteomics, leading to new insights and hypotheses. ESKAPE Act PLUS provides KEGG pathway (4) and GO term (5) activation analyses, which were previously only available for eukaryotic systems and popular model organisms, as well as pathway level gene or protein expression visualizations. An overview of the ESKAPE Act PLUS workflow is provided in Fig. 1. The capabilities, usage, statistical approach, and limitations of ESKAPE Act PLUS are described in detail below, followed by two case studies demonstrating that ESKAPE Act PLUS confirms and extends prior knowledge.

Most existing applications perform gene set enrichment or overrepresentation analysis, but do not predict the activation or repression of biological pathways or functions. The commercially available software tool Ingenuity Pathway Analysis (6) predicts pathway activation or repression and provides pathway level visualizations but does not support prokaryotes. Table 1 compares ESKAPE Act PLUS with other applications that do not require programming knowledge or advanced computational skills. The four existing applications that support prokaryotes, STRING (7), ShinyGO (8), DAVID (9, 10), and ProkSeq (11), are based on overrepresentation analysis, which uses cutoffs and hence does not utilize all data from an experiment, nor do they provide information about activation or repression of biological pathways or functions. Although users of overrepresentation analysis often manually split their data into upregulated or downregulated genes to assess activation and repression, this is inefficient, and it is possible for the same biological function to be simultaneously activated and repressed using this approach.

None of the available tools, including the commercially available tool Ingenuity Pathway Analysis for eukaryotes, offer pathway level gene or protein expression visualizations. For example, STRING provides GO term and KEGG pathway overrepresentation

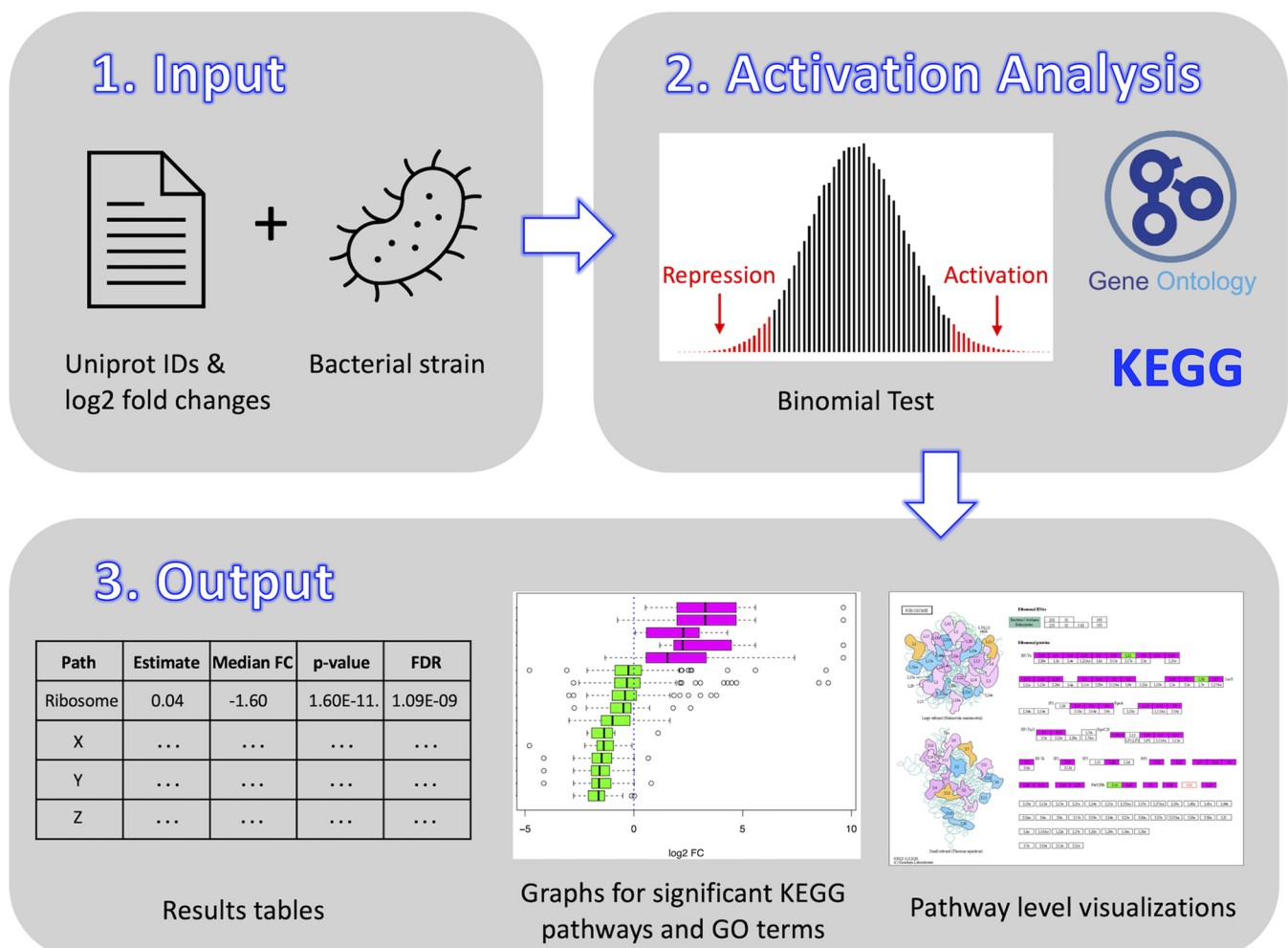

**FIG 1** ESKAPE Act PLUS workflow. An analysis starts with upload of a file containing Uniprot identifiers and log$_2$ fold changes as well as selection of a bacterial strain matching the identifiers in the input file. ESKAPE Act PLUS uses these user-provided inputs as well as KEGG and Gene Ontology (GO) annotations to perform an activation analysis using a binomial test to identify significantly activated or repressed KEGG pathways or GO terms. Outputs include tables with results for all KEGG pathways and GO terms, graphs for significantly activated or repressed KEGG pathways and GO terms, and links to pathway level visualizations of induced and repressed genes or proteins for significant KEGG pathways generated with the KEGG Mapper Color Tool (14, 15). The Gene Ontology logo is available under a CC BY 4.0 license and the KEGG ribosome pathway image is used with permission.

analysis for prokaryotes, but only outputs tables of significant KEGG and GO terms and does not offer graphical output comparing significant terms or pathway level visualizations that highlight differentially expressed genes or proteins. While ShinyGO offers overrepresentation analysis for GO terms and KEGG pathways for eukaryotes, it generally only offers GO term enrichment analysis for prokaryotes. DAVID provides pathway level visualizations, but it does not support many of the clinically relevant strains that are supported by ESKAPE Act PLUS. ProkSeq supports prokaryotes but does not provide pathway level visualizations. Moreover, ProkSeq is designed for the analysis of prokaryotic RNA-seq data starting with fastq files as input, but it does not support proteomics data and, unlike the other tools listed in Table 1, it requires some command line skills.

**ESKAPE Act PLUS features and capabilities.** As shown in Fig. 1, ESKAPE Act PLUS uses biological pathway information from KEGG and GO term annotations as well as user-provided fold changes from bacterial transcriptomics or proteomics experiments to identify pathways and GO terms that are significantly activated or repressed in the experimental condition compared to the reference condition. Application output includes tables with activation analysis statistics for all KEGG pathways and GO terms, graphs for all significantly activated or repressed KEGG pathways and GO terms, as well as visualizations of pathway-level activation or repression for significant KEGG

**TABLE 1** Comparison of ESKAPE Act PLUS to existing applications[a]

| Application | Type | Prokaryotes | Analysis | Visualization | Pathway graphs |
|---|---|---|---|---|---|
| ESKAPE act plus | AA | YES | GO & KEGG | YES | YES |
| STRING (7) | ORA | YES | GO & KEGG | NO | NO |
| ShinyGO (8) | ORA | YES | GO only | YES | NO |
| DAVID (9, 10) | ORA | YES | GO & KEGG | YES | YES |
| ProkSeq (11) | ORA | YES | GO & KEGG | YES | NO |
| WebGestalt (29) | ORA | NO | GO & KEGG | YES | NO |
| GOrilla (30) | ORA | NO | GO only | YES | NO |
| g:Profiler (31) | ORA | NO | GO & KEGG | YES | NO |
| Ingenuity Pathway Analysis (6) | AA | NO | Pathways & Functions | YES | YES |

[a]The first column contains the name and reference for each application, column two describes the type of analysis, activation analysis (AA) or overrepresentation analysis (ORA); column 3 indicates whether analysis of prokaryotic data is supported; column 4 contains available analyses, column 5 shows whether there is a graphical output comparing the significance of KEGG pathways or GO terms in addition to tables; and column 6 indicates the availability of pathway-level visualizations that highlight the differentially expressed genes or proteins provided by the user.

pathways. ESKAPE Act PLUS currently supports 23 strains from 13 species, including the six ESKAPE pathogens as well as several other species that are of relevance to the biomedical research community. These are the most commonly used strains in biomedical high-throughput experiments based on the number of publicly available data sets in repositories such as the Sequence Read Archive (SRA) (12) and the PRoteomics IDEntifications database (PRIDE) (13). A complete list of supported strains is provided in Table 2. ESKAPE Act PLUS calculates activation for KEGG pathways and GO terms that have 4 or more associated genes in the input data provided. Therefore, if the input data contain only three genes on a specific path, no activation score will be calculated. The total number of possible KEGG pathways and GO terms available for each strain are listed in Table 3. All supported bacterial strains except for *Prevotella melaninogenica*, for which there was no publicly available gene or protein expression data, were

**TABLE 2** Overview of supported strains[a]

| Species | Strain | KEGG ID | UniProt ID | Taxon ID |
|---|---|---|---|---|
| *Acinetobacter baumannii* | AYE | aby | ACIBY | 509173 |
| *Acinetobacter baumannii* | MDR-ZJ06 | abz | ACIBM | 497978 |
| *Bacteroides fragilis* | NCTC 9343 | bfs | BACFN | 272559 |
| *Bacteroides ovatus* | 3725 D1 iv | boa | BACOV | 28116 |
| *Bacteroides thetaiotaomicron* | VPI-5482 | bth | BACTN | 226186 |
| *Clostridioides difficile* | 630 | cdf | CLOD6 | 272563 |
| *Clostridioides difficile* | R20291 | cdl | CLODR | 645463 |
| *Enterobacter cloacae* | ATCC 13047 | enc | ENTCC | 716541 |
| *Enterococcus faecium* | DO | efu | ENTFD | 333849 |
| *Escherichia coli* | K-12 MG1655 | eco | ECOLI | 511145 |
| *Escherichia coli* | K-12 W3110 | ecj | ECOLI | 316407 |
| *Escherichia coli* | O157:H7 EDL933 | ece | ECO57 | 155864 |
| *Escherichia coli* | O157:H7 Sakai | ecs | ECO57 | 386585 |
| *Escherichia coli* | BL21(DE3) | ebd | ECOBD | 469008 |
| *Klebsiella pneumoniae* | MGH 78578 | kpn | KLEP7 | 272620 |
| *Prevotella melaninogenica* | ATCC 25845 | pmz | PREMB | 553174 |
| *Pseudomonas aeruginosa* | UCBPP-PA14 | pau | PSEAB | 208963 |
| *Pseudomonas aeruginosa* | PAO1 | pae | PSEAE | 208964 |
| *Staphylococcus aureus* | COL | sac | STAAC | 93062 |
| *Staphylococcus aureus* | USA300 FPR3757 | saa | STAA3 | 367830 |
| *Staphylococcus aureus* | NCTC8325 | sao | STAA8 | 93061 |
| *Staphylococcus aureus* | Newman | sae | STAAE | 426430 |
| *Streptococcus sanguinis* | SK36 | ssa | STRSV | 388919 |

[a]Column 1 contains the species, column 2 the strain, column 3 the KEGG strain identifier, column 4 the Uniprot strain identifier, and column 5 the NCBI Taxonomy identifier. These strains were chosen because they are the most common strains in biomedical high-throughput experiments deposited in public data repositories.

**TABLE 3** Number of available KEGG pathways and GO terms for each strain[a]

| Species | Strain | KEGG paths | GO terms |
|---|---|---|---|
| *Acinetobacter baumannii* | AYE | 92 | 1622 |
| *Acinetobacter baumannii* | MDR-ZJ06 | 93 | 1516 |
| *Bacteroides fragilis* | NCTC 9343 | 85 | 1398 |
| *Bacteroides ovatus* | 3725 D1 iv | 80 | 1285 |
| *Bacteroides thetaiotaomicron* | VPI-5482 | 83 | 1482 |
| *Clostridioides difficile* | 630 | 83 | 1339 |
| *Clostridioides difficile* | R20291 | 84 | 1240 |
| *Enterobacter cloacae* | ATCC 13047 | 94 | 2210 |
| *Enterococcus faecium* | DO | 93 | 2310 |
| *Escherichia coli* | K-12 MG1655 | 94 | 3936 |
| *Escherichia coli* | K-12 W3110 | 94 | 3939 |
| *Escherichia coli* | O157:H7 EDL933 | 94 | 2267 |
| *Escherichia coli* | O157:H7 Sakai | 70 | 1237 |
| *Escherichia coli* | BL21(DE3) | 92 | 2105 |
| *Klebsiella pneumoniae* | MGH 78578 | 99 | 2164 |
| *Prevotella melaninogenica* | ATCC 25845 | 99 | 2929 |
| *Pseudomonas aeruginosa* | UCBPP-PA14 | 101 | 1921 |
| *Pseudomonas aeruginosa* | PAO1 | 71 | 985 |
| *Staphylococcus aureus* | COL | 88 | 1378 |
| *Staphylococcus aureus* | USA300 FPR3757 | 86 | 1357 |
| *Staphylococcus aureus* | NCTC8325 | 85 | 1334 |
| *Staphylococcus aureus* | Newman | 82 | 1776 |
| *Streptococcus sanguinis* | SK36 | 70 | 1191 |

[a]Column 1 contains the species, column 2 the strain, column 3 the number of available KEGG pathways, and column 4 the number of available GO terms.

validated using existing data. The data sources and results of the validation are listed in Table S1, and the contents of the table are described in detail in the Material and Methods section.

**ESKAPE Act PLUS usage.** The ESKAPE Act PLUS web application can be freely accessed at http://scangeo.dartmouth.edu/ESKAPE/. To start an activation analysis, users upload a file in comma separated values (.csv) format with their input data in two columns without headers. The first column of the csv file must contain UniProt identifiers, and the second column must contain $\log_2$ fold changes for the UniProt identifiers specified in Column 1 for the comparison of interest. Column 2 must contain numeric values only, no text, and may not contain any missing values. The application provides diagnostic error messages if an input file does not meet these specifications. An example input data file for *Pseudomonas aeruginosa* strain PA14 can be downloaded via the application interface to illustrate the correct data format and serve as an input file for users who would like to try the application but are lacking a suitable input file. Step-by-step instructions on how to convert gene identifiers to UniProt identifiers using the UniProt ID conversion tool are provided in the User Manual that is downloadable from the application interface. ESKAPE Act PLUS generates two kinds of output – in-application output and downloadable output. Following an activation analysis, the in-app output appears in tabs in the main panel of the application, while downloadable output becomes available via the "Download results" button that appears in the left control panel of the app. In-app output includes the number of user-provided UniProt identifiers that mapped to the internal KEGG and GO database, the number of significant KEGG pathways and GO terms, as well as box plots and tables for all significant (FDR < 0.05) KEGG pathways and GO terms. Downloadable output is comprised of box plots for all significant KEGG pathways and GO terms, tables with statistical results for all KEGG pathways and GO terms that were tested, and optional KEGG Mapper (14, 15) generated images for significantly activated or repressed KEGG pathways, if applicable. ESKAPE Act PLUS also seamlessly interfaces with output from CF-Seq (16), a web application that facilitates re-analysis of publicly

available RNA-seq data relevant to cystic fibrosis research, including transcriptomic data from bacteria that are common causes of lung infection.

**Statistical Approach.** ESKAPE Act PLUS uses a binomial test to assess whether a given KEGG pathway or GO term contains significantly more induced or repressed genes or proteins than would be expected by chance. Under the null hypothesis, about 50% of the genes or proteins associated with any KEGG pathway or GO term would be expected to respond to treatment with a FC $> 0$, while the other 50% would respond with a FC $< 0$. Statistically significant divergence from this 50:50 split is assessed using binomial tests based on the fold changes of all genes or proteins associated with a KEGG pathway or GO term. The reference set for the binomial test is all genes or proteins from the input file (rather than all genes or proteins known for a given species). This choice corrects for a potential bias that would be introduced by any experimental design that tends to only detect certain kinds of genes or proteins. Only KEGG pathways or GO terms with an FDR-corrected $P$-value $< 0.05$ are considered significantly induced or repressed and displayed in figures and tables of the in-app results output, whereas the downloadable csv files contain the results for all tested KEGG pathways and GO terms.

**Case study 1–*Enterococcus faecium* proteomic data.** We re-analyzed publicly available data (17) to demonstrate that ESKAPE Act PLUS confirms prior results and that it can be used to re-analyze publicly available data to gain additional insights and form new hypotheses. ESKAPE Act PLUS was used to analyze fold changes in proteomic data from extracellular vesicles secreted by *Enterococcus faecium* strain DO during the exponential growth phase compared to extracellular vesicles secreted during stationary phase. The authors of the original publication performed functional enrichment analysis and identified 10 functional annotation clusters in vesicle associated proteins from exponential growth conditions (17) (Fig. 2A). ESKAPE Act PLUS confirmed all 10 biological functions from the original manuscript while identifying an additional 3 significant GO terms and 5 significant KEGG pathways (Fig. 2B and C). In the case of proteins associated with the ribosome, which was the biological function with the largest enrichment score in the original publication (Fig. 2A), ESKAPE Act PLUS calculated significant activation overall due to a larger number of upregulated versus repressed proteins. Yet even though there were many differentially expressed ribosomal proteins, fold changes for upregulated and downregulated proteins canceled each other out, resulting in a net fold change close to zero (Fig. 2C). This example illustrates that despite a high enrichment score due to many differentially expressed proteins associated with a biological function, if some of the proteins are upregulated while others are downregulated, the overall biological effect on a pathway or function may be neither activation nor repression. Thus, performing activation analysis rather than enrichment analysis adds valuable information that improves interpretation. Moreover, ESKAPE Act PLUS identified additional significant GO terms and KEGG pathways that were not mentioned in the original publication (marked with red asterisks in Fig. 2B and C), extending the findings reported in the literature and providing the basis for follow-up experiments. For example, the GO term with the largest median negative fold change identified by ESKAPE Act PLUS, zinc ion binding, as well as the significantly downregulated KEGG pathways Bacterial secretion system, Quorum sensing, and Oxidative phosphorylation were not identified in the original publication. Moreover, the KEGG pathway activation analysis performed by ESKAPE Act PLUS provides more detail on which metabolic pathways were particularly affected (Fig. 2C), rather than just listing the very general GO term "Metabolism".

**Case study 2–*Clostridioides difficile* transcriptomic data.** Previous work by Boekhoud et al. describes the role of alternative sigma factor sigma B in protecting *C. difficile* strain 630 against stressors such as reactive oxygen species (ROS) and antimicrobial compounds (18). The authors of the original study found that most genes downregulated upon overexpression of sigma B were associated with flagellar motility. Re-analysis of the transcriptomic data (fold changes for a strain of *C. difficile* that overexpresses sigma B compared to a control strain) with ESKAPE Act PLUS confirmed the

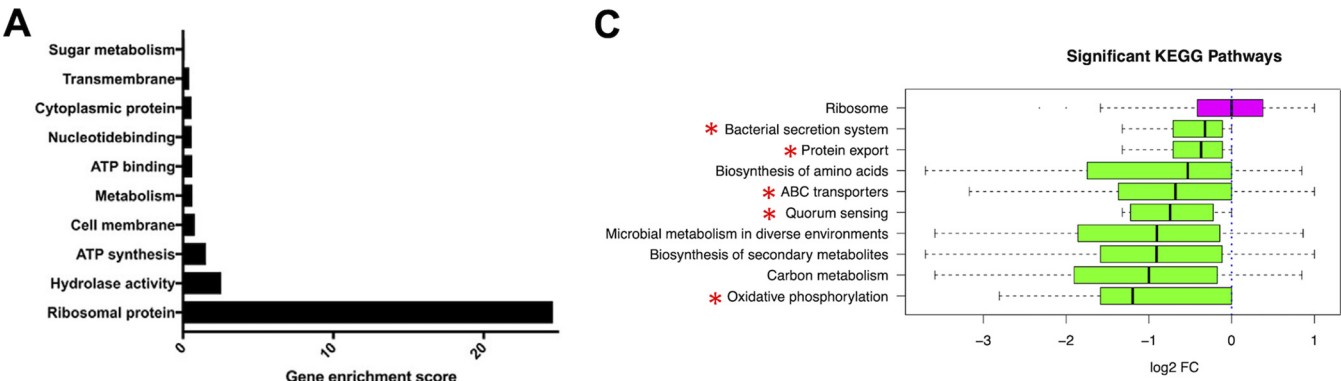

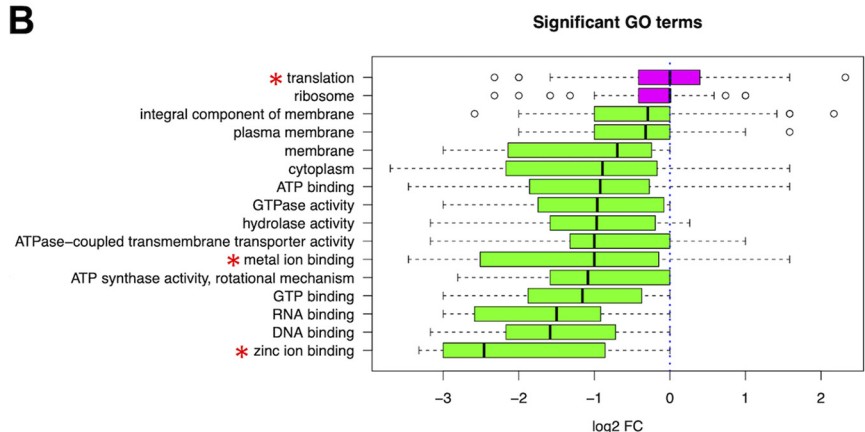

**FIG 2** Case Study 1 - *Enterococcus faecium* proteomic data. (A) Results of a gene ontology enrichment analysis for proteomic data from extracellular vesicles secreted by *Enterococcus faecium* strain DO during the exponential versus stationary growth phase performed by Wagner et al. (17). This figure was originally published by Wagner et al. (17) as Fig. 6A and is licensed under CC BY-NC-ND 4.0. (B) Significantly activated (magenta) or repressed (green) GO terms identified by ESKAPE Act PLUS. Red asterisks highlight GO terms that were not mentioned in the original publication. (C) Significantly activated (magenta) or repressed (green) KEGG pathways identified by ESKAPE Act PLUS. Red asterisks highlight KEGG pathways that were not mentioned in the original publication.

KEGG pathway "Flagellar Assembly" (Fig. 3A) as the most significantly repressed pathway (FDR = $1*10^{-5}$) with a median $\log_2$ fold change of -1.41 across all genes on the path. In addition, ESKAPE Act Plus extends the previously published findings by identifying Bacterial Chemotaxis (Fig. 3B) as the second most significantly repressed pathway (FDR = 0.003, median $\log_2$ fold change = -1.12). This is a new insight, as the original publication did not mention that bacterial chemotaxis may be affected by overexpression of sigma B. In summary, ESKAPE Act PLUS is a user-friendly web application that not only confirms previous findings but extends beyond known observations to generate new knowledge and hypotheses that can be validated in wet-lab experiments.

**Limitations.** To identify the largest possible number of significantly activated or repressed pathways and GO terms, an ESKAPE Act PLUS user would need to provide fold change information for all genes or proteins that have functional annotations. While this might be possible in the case of RNA-seq studies, proteomics experiments only detect more abundant proteins, typically accounting for 20–40% of all proteins. ESKAPE Act PLUS may fail to identify significant KEGG pathways or GO terms if fold change information is available for fewer than a few hundred genes or proteins. Another limitation is that the application cannot interpret changes in genes of unknown function that are not associated with a KEGG pathway or GO term. Finally, ESKAPE Act PLUS uses a simplified view of a pathway such that a pathway is considered significantly activated or repressed if more genes on the path are activated or repressed than would be expected by chance. This simplified view assumes that when a pathway is activated, all genes or proteins on the pathway will be induced. However, this is not always the case, for example when a pathway is turned on, upstream

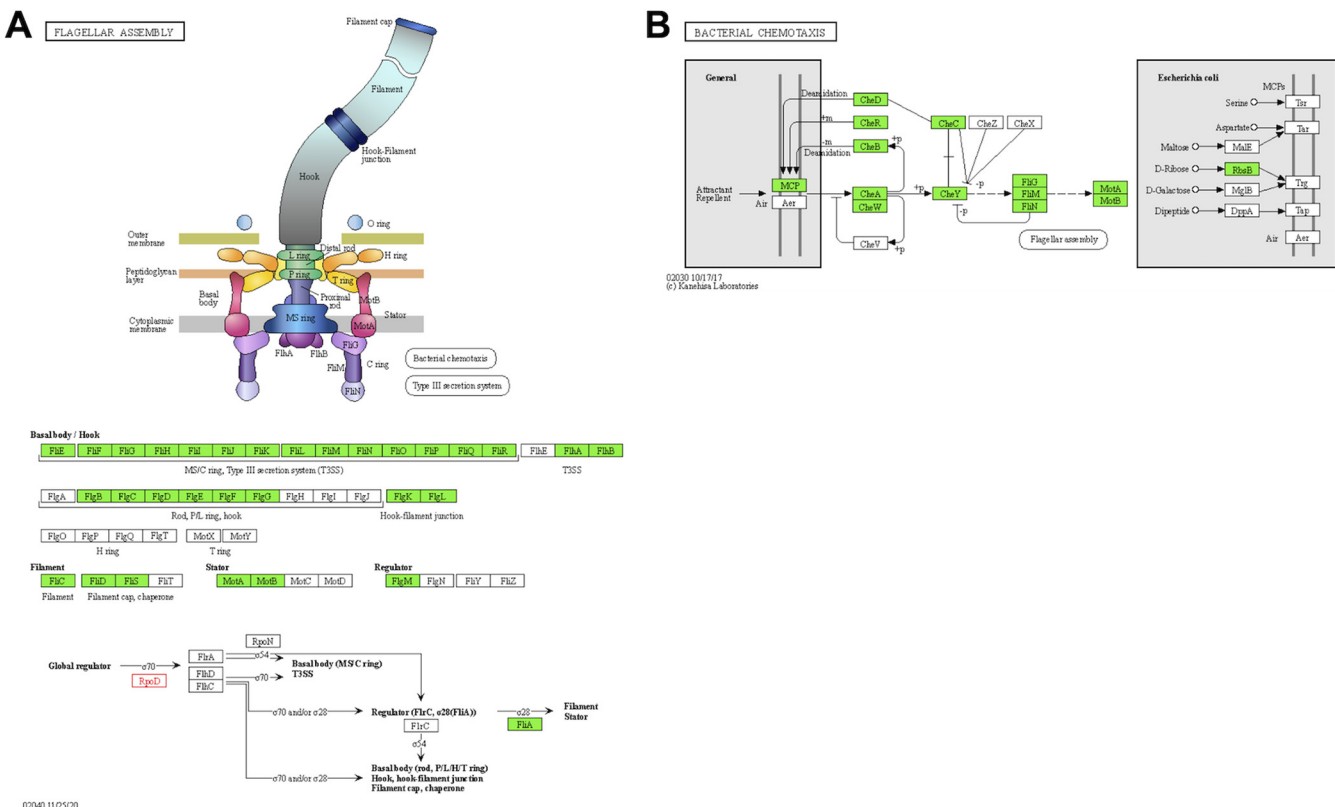

**FIG 3** Case Study 2 - *Clostridioides difficile* transcriptomic data. Results of the ESKAPE Act PLUS re-analysis of transcriptomic data for *C. difficile* strain 630 overexpressing alternative sigma factor sigma B compared to a control strain, originally published by Boekhoud et al. (18). (A) ESKAPE Act PLUS identified the KEGG pathway "Flagellar Assembly" as the most significantly repressed pathway, confirming the finding of the original study that overexpression of sigma B results in repression of genes associated with flagellar motility. (B) The KEGG pathway "Bacterial Chemotaxis" was identified by ESKAPE Act PLUS as the second most significantly repressed pathway, thus generating new knowledge beyond what was reported in the original publication of the data. Genes with a negative fold change are highlighted in green and genes that were not detected in the experiment are depicted in white. KEGG pathway images were generated with the KEGG Mapper Color Tool (14, 15) and are used with permission.

regulators that act as repressors are often turned off. Unlike Ingenuity Pathway Analysis, a commercial tool that is human-curated, ESKAPE Act PLUS does not account for more complicated pathway relationships such as negative regulators. Nonetheless, in practice the simplifying assumption used by ESKAPE Act PLUS has minimal impact on its sensitivity, because a small number of negative regulators with an inverse fold change from the rest of the genes or proteins on a pathway won't prevent the pathway from reaching statistical significance.

A final limitation of our system is that visualizations may not perfectly represent the results of the binomial test because the statistical analysis uses species specific annotations, but the pathway level visualization uses canonical KEGG pathways.

**Conclusions.** ESKAPE Act PLUS provides high-quality statistics and graphical representations not available using other web-based systems to assess whether prokaryotic biological functions are activated or repressed by experimental conditions. While there are tools such as the commercially available Ingenuity Pathway Analysis that perform pathway activation analysis for humans and other eukaryotic model organisms, to our knowledge ESKAPE Act PLUS is the first application that extends this functionality to prokaryotes.

## MATERIALS AND METHODS

**Annotations.** ESKAPE Act PLUS uses biological pathway information from the Kyoto Encyclopedia of Genes and Genomes (KEGG) (4), obtained through the KEGGREST R package version 1.32.0 (23), and gene ontology (GO) term annotations (5, 24) for biological processes, molecular functions, and cellular components retrieved from the Universal Protein Resource (UniProt) (25).

**Activation analysis.** ESKAPE Act PLUS uses a binomial test to assess whether there is a significant number of upregulated or downregulated genes or proteins among all genes or proteins detected for a

given KEGG pathway or GO term. KEGG pathway and GO term activation analysis were performed as previously described (26–28). The universe for the binomial test is all genes or proteins from the input file (rather than all genes or proteins known for a given species). Using identifiers found in the source file as the reference avoids the possibility of introducing biases related to tissue type and sample preparation. Pathway enrichment $P$-values from the binomial test are FDR-corrected to account for the total number of KEGG pathways or GO terms that were tested. Only KEGG pathways or GO terms with an FDR-corrected $P$-value $< 0.05$ are considered significantly induced or repressed and displayed in figures and tables of the in-app results output. Results tables also contain estimates for activation or repression (with values less than 0.5 representing repression of a given KEGG pathway or GO term, while values above 0.5 indicate activation) as well as median fold changes and uncorrected $P$-values. While the in-app output only displays results for significant KEGG pathways and GO terms, downloadable csv files contain the results for all tested KEGG pathways and GO terms to allow users to identify borderline cases with uncorrected $P$-values $< 0.05$ that just barely missed the significance cutoff of FDR $< 0.05$.

**Pathway level visualizations.** KEGG Mapper (14, 15) provides a color mapping tool that ESKAPE Act PLUS uses to generate KEGG pathway images overlaid with color information about gene or protein level fold changes, allowing for easy visualization of pathway-level activation or repression. Genes or proteins with a positive fold change are highlighted in magenta, while genes or proteins with a negative fold change are depicted in green. Genes or proteins that were not detected in the experiment are shown in white. ESKAPE Act PLUS uses an API interface with KEGG Mapper's Color Tool to generate links to downloadable pathway level images for all significant KEGG pathways. These clickable links are provided in the KEGG pathway results table inside of the application as well as in a downloadable html document for future reference.

**Validation of bacterial strains.** Proper function of ESKAPE Act PLUS for all supported bacterial strains was validated using existing data, except for *Prevotella melaninogenica*, for which there was no publicly available gene or protein expression data. A summary table of the strain validation is provided as Table S1. Columns 1 and 2 contain the species and strain, respectively, column 3 provides the source of the test data, and columns 4 and 5 list the percentage of available KEGG or GO identifiers that matched to the user-provided input IDs. In general, proteomics studies tend to have a lower percentage of matched IDs because only a subset of more abundant proteins is detected in a typical proteomics experiment, whereas transcriptomic studies in prokaryotes tend to detect most if not all possible genes. Columns 6 and 7 contain the number of significant KEGG pathways and GO terms for each analysis, and column 8 provides information on whether the results of ESKAPE Act PLUS agree with previously published data (when available). This comparison is not applicable for studies with no significant KEGG pathways or GO terms (N/A) or those that do not have an associated publication. Except for one of the studies, all published studies for which ESKAPE Act PLUS identified significant KEGG pathways or GO terms confirmed previously published results and extended it in many of the cases. The DOI link to the original publication for all published studies is provided in column 9.

**Case studies.** To demonstrate that ESKAPE Act PLUS confirms and extends prior knowledge, we re-analyzed existing data from two publications (17, 18). The study by Wagner et al. (17) included proteomic data from extracellular vesicles secreted by *Enterococcus faecium* strain DO during the exponential and stationary growth phase. $Log_2$ fold changes for the comparison between vesicles from the exponential and stationary growth phase were calculated using the unique peptide counts provided as a supplementary table in the original study. These $log_2$ fold changes as well as the matching Uniprot identifiers were used to generate an input file for ESKAPE Act PLUS for Case Study 1. For Case Study 2, transcriptomic data from *Clostridioides difficile* strain 630 overexpressing alternative sigma factor sigma B compared to a control, originally published by Boekhoud et al. (18), was reanalyzed with ESKAPE Act PLUS. A supplementary file with fold change information (GSE152515_Ptet-sigB_vs_Ptet-slucopt_hyb2_induced_with_thiamphenicol.xls.gz) was downloaded from NCBI GEO Series GSE152515. Gene identifiers were converted to Uniprot Entry IDs using the Uniprot ID conversion tool, as explained in detail in the ESKAPE Act PLUS User Guide.

**Data availability.** The source code for ESKAPE Act PLUS is written in the R software environment for statistical computing and graphics version 4.1.1 (19). The code as well as all files required to run the application can be obtained from a Github repository (https://github.com/StantonLabDartmouth/ESKAPE_Act). ESKAPE Act PLUS is implemented as an R Shiny web application (20) using the R packages shiny (21) version 1.7.1 and shinyjs (22) version 2.1.0. The web application can be accessed freely at http://scangeo .dartmouth.edu/ESKAPE/.

## SUPPLEMENTAL MATERIAL

Supplemental material is available online only.
**TABLE S1**, XLSX file, 0.01 MB.

## ACKNOWLEDGMENTS

We thank Benjamin Ross and Paige Salerno for providing data to validate *Bacteroides fragilis* and *Bacteroides ovatus*.

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
