## [Reviewer comments · mSystems]

ESKAPE Act PLUS: Pathway Activation Analysis for Bacterial Pathogens

Katja Koeppen, Thomas Hampton, Samuel Neff, and Bruce Stanton

Corresponding Author(s): Katja Koeppen, Geisel School of Medicine at Dartmouth

Review Timeline:

Submission Date:	May 18, 2022
Editorial Decision:	July 22, 2022
Revision Received:	September 13, 2022
Accepted:	September 28, 2022

Editor: Ryan McClure

Reviewer(s): Disclosure of reviewer identity is with reference to reviewer comments included in decision letter(s). The following individuals involved in review of your submission have agreed to reveal their identity: Francois Lebreton (Reviewer #5)

Transaction Report:

DOI: <https://doi.org/10.1128/msystems.00468-22>

July 22, 2022

Dr. Katja Koeppen
Geisel School of Medicine at Dartmouth
Hanover

Re: mSystems00468-22 (ESKAPE Act PLUS: Pathway Activation Analysis for Bacterial Pathogens)

Dear Dr. Katja Koeppen:

Thank you for submitting your manuscript to mSystems. We have completed our review and I am pleased to inform you that, in principle, we expect to accept it for publication in mSystems. However, acceptance will not be final until you have adequately addressed the reviewer comments.

Preparing Revision Guidelines

Sincerely,

Ryan McClure

Editor, mSystems

Journals Department
Reviewer comments:

Reviewer #1 (Comments for the Author):

Overall, this is a very well written manuscript. In addition, the website for ESKAPE ACT Plus, the sample data set provided, and the user manual that can be found on the website were all helpful. I was successfully able to use the sample data set to evaluate the functionality of the web tool and found the overall experience to be clear and easy to use.

I do have one suggestion, although this may not be possible for technical reasons of the website. I do like the figures that can be generated using the webtool, but the image quality is not high quality enough that it would be appropriate for publication. Is it possible to include an option for downloading a higher resolution PNG image or a vector file?

Reviewer #3 (Comments for the Author):

The integration of technology and computer science in the scientific community is very necessary and exhibits an important factor in the advancement of the field. The developed web application is an excellent example of the integration of such. The simplicity of the representation and model is very crucial, particularly to the bacterial pathogens which are very prone to AMR. The public issue of AMR is a global problem which requires attention and subsequent approaches to combat AMR.

Reviewer #5 (Comments for the Author):

This work by Koeppen et al aims at providing the microbial scientific community with an easy-to-use tool to perform pathway activation analysis (and visualization) using omics expression data. While the tool is already available online at no cost, this short report i) documents the methodology, function, and output of ESKAPE Act PLUS, ii) provides a proof of concept/superiority by re-analyzing a publicly available dataset and, iii) briefly compares strengths and weaknesses to existing tools. The manuscript is clearly written (as is the user manual available on the webpage) and such tools are clearly needed but, in this reviewer's opinion, a number of limitations exist and might limit the impact and future use. This is further detailed in the specific remarks below.

Major remarks:

1. While likely accurate, the statement that the species/strains for which ESKAPE Act PLUS is available are "the most commonly used strains in biomedical high-throughput experiments" is not substantiated by reference 9 or by quantitative data. More importantly, in the genomic era, the availability of only 23 strains might be an important limitation for the wide use of this tool by the community. At the time of this review, pathways for >7,000 bacterial strains are available in the KEGG database. What limits the integration of those into ESKAPE Act PLUS? are there plans to expand to more strains/species in the future?
2. The description of ESKAPE Act PLUS is well detailed in the "features and capabilities" section of the main text but it is somehow underwhelming in the figures and visuals. This is particularly notable as ESKAPE Act PLUS is offered as a visualization tool. As an example, the pathway level visualization/graphs (which is a clear, defining feature of ESKAPE Act PLUS) is only displayed as a small snapshot, low-resolution snapshot on Figure 1 and details are not readable.
3. Expanding on point #2, the proof-of-concept using the Enterococcus dataset is interesting but might not have been the best choice to display the power of ESKAPE Act PLUS. A genome-wide dataset (similar to the one available for PA14 on the website) with notable differences observable on pathways/maps would better illustrate the strengths of the tools and attract the attention of future users.
4. The comparison to existing tools is a key aspect of this publication. While capabilities are summarized in table 1, the performance of the tools are not compared and contrasted using a common dataset which, in this reviewer's opinion, limits the ability of a reader to make an informed decision. Finally, other tools like DAVID and ProkSeq have either been widely used (DAVID) or provide a complete analytical pipeline from the fastq reads to pathways visuals (ProkSeq) and should likely be discussed in this manuscript.

Minor remarks:

1. Please add line numbering to the draft manuscript to facilitate referencing during the review process.
2. Page 3, line 3 of introduction. Revise "medial" to "medical"
3. Page 3, line 10 of introduction. Should this read "species" instead of strains. Or maybe XX strains from YY species.

Title: ESKAPE Act Plus: Pathway Activation Analysis for Bacterial Pathogens

Comments: The integration of technology and computer science in the scientific community is very necessary and exhibits as an important factor in the advancement of the field. The developed web application is an excellent example of the integration of such. The simplicity of the representation and model is very crucial particularly to the pathogens which are very prone to AMR. The public issue of AMR is a global problem which requires attention and subsequent approaches to combat AMR.

1. The clinical relevance and focus of web application ESKAPE Act Plus are described
2. The introduction elaborated on the ESKAPE Act Plus web application adequately
3. Results and discussion described the comparison of ESKAPE Act Plus to existing tools, shedding light on the similarities and difference of the tools and highlighting features of ESKAPE Act Plus appropriately
 - a. Case study included demonstrates the functionality of the web application clearly
 - b. Tables and figures used aid in visual representation which are sufficiently portrayed
4. Limitations are evaluated systematically which is necessary to allow a broader image of the qualities of the web application
5. Conclusion is clear and concise summary

Response to Reviewer Comments

Reviewer comments:

Reviewer #1 (Comments for the Author):

Overall, this is a very well written manuscript. In addition, the website for ESKAPE ACT Plus, the sample data set provided, and the user manual that can be found on the website were all helpful. I was successfully able to use the sample data set to evaluate the functionality of the web tool and found the overall experience to be clear and easy to use.

I do have one suggestion, although this may not be possible for technical reasons of the website. I do like the figures that can be generated using the webtool, but the image quality is not high quality enough that it would be appropriate for publication. Is it possible to include an option for downloading a higher resolution PNG image or a vector file?

Response: The overview graphs for significantly activated or repressed KEGG pathways or GO terms are high resolution publication quality pdf files. The individual KEGG pathway images are png files, but unfortunately, their resolution is limited by the original source KEGG graphs, which are raster images.

Reviewer #3 (Comments for the Author):

The integration of technology and computer science in the scientific community is very necessary and exhibits an important factor in the advancement of the field. The developed web application is an excellent example of the integration of such. The simplicity of the representation and model is very crucial, particularly to the bacterial pathogens which are very prone to AMR. The public issue of AMR is a global problem which requires attention and subsequent approaches to combat AMR.

Reviewer #5 (Comments for the Author):

This work by Koeppen et al aims at providing the microbial scientific community with an easy-to-use tool to perform pathway activation analysis (and visualization) using omics expression data. While the tool is already available online at no cost, this short report i) documents the methodology, function, and output of ESKAPE Act PLUS, ii) provides a proof of concept/superiority by re-analyzing a publicly available dataset and, iii) briefly compares strengths and weaknesses to existing tools. The manuscript is clearly written (as is the user manual available on the webpage) and such tools are clearly needed but, in this reviewer's opinion, a number of limitations exist and might limit the impact and future use. This is further detailed in the specific remarks below.

Major remarks:

1. While likely accurate, the statement that the species/strains for which ESKAPE Act PLUS is available are "the most commonly used strains in biomedical high-throughput experiments" is not substantiated by reference 9 or by quantitative data. More importantly, in the genomic era, the availability of only 23 strains might be an important limitation for the wide use of this tool by the community. At the time of this review, pathways for >7,000 bacterial strains are available in the KEGG database. What limits the integration of those into ESKAPE Act PLUS? are there plans to expand to more strains/species in the future?

Response: The strains currently supported by ESKAPE Act PLUS are based on those strains that are most frequently used in research experiments as determined by transcriptomics data deposited in Sequence Read Archive (SRA) as well as proteomics data deposited in the PRoteomics IDentifications Database (PRIDE). A statement explaining the strain selection can be found on lines 136-138 of the manuscript.

A major limitation to the number of supported strains is that ensuring correct annotation and proper functionality is a manual process that does not scale to hundreds or thousands of strains. We are planning to add additional strains in the future upon user request.

2. The description of ESKAPE Act PLUS is well detailed in the "features and capabilities" section of the main text but it is somehow underwhelming in the figures and visuals. This is particularly notable as ESKAPE Act PLUS is offered as a visualization tool. As an example, the pathway level visualization/graphs (which is a clear, defining feature of ESKAPE Act PLUS) is only displayed as a small snapshot, low-resolution snapshot on Figure 1 and details are not readable.

Response: The intention of Figure 1 is to provide an overview of the application and its outputs. Figure 2 includes larger, high-resolution images of summary graphs for significant KEGG pathways and GO terms and in the new version of the manuscript, we have included a second case study (lines 220-234) and a new figure (Fig. 3) that showcases the pathway level output. Due to copyright restrictions, we cannot use pathview output images in a publication. Therefore, we have updated ESKAPE Act PLUS to use KEGG Mapper instead of pathview to generate improved pathway level images and we have received permission to use the resulting images in our publication. KEGG Mapper has the additional benefit of providing a more user-friendly experience and much shorter processing time compared to pathview. In the new version of the manuscript, we have replaced the methods section about pathview with a description of and references to KEGG Mapper (lines 301-310).

3. Expanding on point #2, the proof-of-concept using the Enterococcus dataset is interesting but might not have been the best choice to display the power of ESKAPE Act PLUS. A genome-wide dataset (similar to the one available for PA14 on the website) with notable differences observable on pathways/maps would better illustrate the strengths of the tools and attract the attention of future users.

Response: We have added an additional genome-wide transcriptomic dataset for *Clostridium difficile* as a second case study to the new version of the manuscript (lines 220-234) and generated an additional figure (Fig. 3) to illustrate the pathway level output. In the original publication for this dataset (Boekhoud et al., 2020; <https://journals.asm.org/doi/10.1128/mSphere.00728-20>), the authors noted that "the majority of genes downregulated upon overexpression of sigmaB fall into a single functional group (flagellar motility)". This finding was confirmed by ESKAPE Act PLUS, which identified the KEGG pathway "Flagellar Assembly" as the most significantly repressed pathway (FDR = 1×10^{-5}) with a median log2 fold change of -1.41 across all genes on the path. In addition, ESKAPE Act Plus extends the previously published findings by identifying Bacterial Chemotaxis as the second most significantly repressed pathway (FDR = 0.003, median log2 fold change = -1.12). The pathway level output images for Flagellar Assembly and Bacterial Chemotaxis generated by KEGG Mapper have been included as Figure 3 in the revised manuscript.

4. The comparison to existing tools is a key aspect of this publication. While capabilities are summarized in table 1, the performance of the tools are not compared and contrasted using a common dataset which, in this reviewer's opinion, limits the ability of a reader to make an informed decision. Finally, other tools like DAVID and ProkSeq have either been widely used (DAVID) or provide a complete analytical pipeline from the fastq reads to pathways visuals (ProkSeq) and should likely be discussed in this manuscript.

Response: We agree with the reviewer that the comparison of existing tools is a key aspect of this publication and we have therefore sharpened the section on lines 93-125, which now discusses ProkSeq and DAVID directly, and we have added these tools to Table 1. As the reviewer implies, every tool has strengths and weaknesses. We have clarified that while ESKAPE Act PLUS does not handle fastq files as input (as ProkSeq does), unlike ProkSeq, ESKAPE Act PLUS does not require any command line skills, it offers a measure of activation or repression of paths, not just overrepresentation of differentially expressed genes, and it covers certain popular strains (e.g. PA14) that DAVID does not support.

Minor remarks:

1. Please add line numbering to the draft manuscript to facilitate referencing during the review process.

Response: Line numbering has been added to the revised manuscript.

2. Page 3, line 3 of introduction. Revise "medial" to "medical"

Response: We have corrected this in the new version of the manuscript (line 58).

3. Page 3, line 10 of introduction. Should this read "species" instead of strains. Or maybe XX strains from YY species.

Response: In the new version of the manuscript, we have changed this sentence to "ESKAPE Act PLUS currently supports analysis of 23 strains of bacteria from 13 species [...]" (lines 45-46).

September 28, 2022

Dr. Katja Koeppen
Geisel School of Medicine at Dartmouth
Hanover

Re: mSystems00468-22R1 (ESKAPE Act PLUS: Pathway Activation Analysis for Bacterial Pathogens)

Dear Dr. Katja Koeppen:

Your manuscript has been accepted, and I am forwarding it to the ASM Journals Department for publication. For your reference, ASM Journals' address is given below. Before it can be scheduled for publication, your manuscript will be checked by the mSystems production staff to make sure that all elements meet the technical requirements for publication. They will contact you if anything needs to be revised before copyediting and production can begin. Otherwise, you will be notified when your proofs are ready to be viewed.

Publication Fees:

If you would like to submit a potential Featured Image, please email a file and a short legend to mSystems@asmusa.org. Please note that we can only consider images that (i) the authors created or own and (ii) have not been previously published. By submitting, you agree that the image can be used under the same terms as the published article. File requirements: square dimensions (4" x 4"), 300 dpi resolution, RGB colorspace, TIF file format.

We recognize that the video files can become quite large, and so to avoid quality loss ASM suggests sending the video file via <https://www.wetransfer.com/>. When you have a final version of the video and the still ready to share, please send it to mSystems staff at mSystems@asmusa.org.

Sincerely,

Ryan McClure
Editor, mSystems

Journals Department
Table S1: Accept